# HPV-Associated Sexually Transmitted Infections in Cervical Cancer Screening: A Prospective Cohort Study

**DOI:** 10.3390/v17020247

**Published:** 2025-02-11

**Authors:** Miriam Latorre-Millán, Alexander Tristancho-Baró, Natalia Burillo, Mónica Ariza, Ana María Milagro, Pilar Abad, Laura Baquedano, Amparo Borque, Antonio Rezusta

**Affiliations:** 1Research Group on Infections Difficult to Diagnose and Treat, Institute for Health Research Aragón, Miguel Servet University Hospital, 50009 Zaragoza, Spain; aitristancho@salud.aragon.es (A.T.-B.); nburillon@salud.aragon.es (N.B.); mparizas@salud.aragon.es (M.A.); amilagro@salud.aragon.es (A.M.M.); mpa.alejaldre@gmail.com (P.A.); arezusta@salud.aragon.es (A.R.); 2Gynaecology Department, Miguel Servet University Hospital, 50009 Zaragoza, Spain; lbaquedano@salud.aragon.es (L.B.); aborqueib@salud.aragon.es (A.B.)

**Keywords:** sexually transmitted infections, human papillomavirus, cervical cancer screening

## Abstract

High-risk human papillomavirus (HR-HPV) and other sexually transmitted infections (STIs-O) are promoters to the development of cervical cancer (CC), especially when they co-exist. This study aims to determine the prevalence of the major STIs-O and the rate of co-infection in women previously diagnosed with HR-HPV infection. For this observational study, 254 women aged 25–65 years who were being followed up for HR-HPV infection (without a CC history) were recruited at a hospital’s Gynaecology Department from February 2024 to November 2024. Their endocervical specimens were collected and processed for HR-HPV, *Chlamydia trachomatis*, *Neisseria gonorrhoeae*, *Mycoplasma genitalium*, and *Trichomonas vaginalis* detection by RT-PCR using commercially available reagents and equipment. The overall rate of infection was 38.6% for HPV and 4.3% for ITSs-O (3.8% in HPV-negative women and 5.1% in HPV-positive women). The presence of ITSs-O in women aged 25–34 was higher in those with a persistent positive result for HR-HPV (20.0% vs. 4.2%). Diverse multiple co-infections were found in HPV-positive women, whilst some single STIs-O were found in HPV-negative women. These results support the benefits of STI-O screening beyond an HR-HPV positive result, especially in those women under 35 years old.

## 1. Introduction

Cervical cancer (CC) represents the second most prevalent oncological disease among women worldwide, with a significant impact on quality of life and economic costs. In Spain, estimates for 2023 indicate that 2047 women were diagnosed with CC and 664 deaths were attributed to this disease [1]. Nevertheless, there is substantial scientific evidence that preventive measures against CC can be both feasible and effective. Firstly, although not typically fatal, the disease predominantly affects young, sexually active women. Secondly, it is associated with modifiable environmental risk factors. Thirdly, it is frequently preceded by progressive cell dysplasia. Finally, the cervix is easily accessible for diagnostic procedures, specimen collection, and indirect observation.

In relation to the aforementioned environmental risk factors, a number of microorganisms have been identified as contributing to the development of alterations and malignancies that lead to CC by altering the cellular environment [2]. However, the Spanish CC screening programme, apart from smear tests, only include high-risk human papillomavirus (HR-HPV) detection [3,4,5], despite the fact that HPV infections are encountered by over 80% of women during their lifetime, with less than 2% developing CC [6] and the majority of infections being cleared by the host [7,8].

In fact, the risk of developing CC and/or cervical intraepithelial neoplasia (CIN) as well as the incidence and persistence of HR-HPV and other sexually transmitted infections (STI-O) have been found to be increased in the presence of vaginal dysbiosis and vaginosis (symptomatic vaginal dysbiosis) [9,10,11]. Conversely, the absence of STI-O in addition to HR-HPV is associated with a reduced likelihood of histological abnormalities. Furthermore, certain microorganisms have been found to be associated with the appearance of CIN even with a higher OR for CC than HR-HPV. This is exemplified by *Chlamydia trachomatis* (CT), the most prevalent STI in western countries: its co-infection with HR-HPV increases the OR for high-grade squamous intraepithelial lesion (HGSIL) and CC even further [10,12]. Indeed, screening for CT is recommended in sexually active women under 25 years of age and in sexually active women over 25 years old when at increased risk, such as those having new partners, multiple partners, or a partner who has an STI [13].

Moreover, in addition to CC, STIs can impact personal wellbeing, mental health, and relationships and lead to significant complications “including pelvic inflammatory disease, ectopic pregnancy, postpartum endometriosis, infertility, and chronic abdominal pain in women; adverse pregnancy outcomes, including abortion, intrauterine death, and premature delivery; neonatal and infant infections and blindness; urethral strictures and epididymitis in men” and other extra-gynaecological malignancies depending on the specific pathogen involved, such as arthritis secondary to gonorrhoea and chlamydia [14]. Therefore, it is of the utmost importance that diagnosis and treatment be promptly initiated in order to ensure optimal medical care [13,14]. The greatest risk factor for an STI is a previous diagnosis of another STI. This is due in part to the fact that the subsequent oxidative stress and/or chronic inflammation serves as a facilitator for new pathogens to exert damage, thereby increasing the risk of co-infections, chronic infections, and further disorders. Despite the focus on HR-HPV infection’s role in the development of CC, it is crucial to recognise that HPV is the most frequent STI worldwide and should be treated as such. This entails assessing for co-infection, establishing a window of opportunity for treatment and promoting HPV clearance.

There is considerable variation in the prevalence of STIs-O between countries; however, it is higher in women who are HR-HPV positive. Hence, numerous studies have emphasised the necessity for routine screening and monitoring of HR-HPV co-infection with STIs-O [15,16,17,18], aligning with the viewpoint that this approach is cost-effective when the rate of co-infection exceeds 3% [19,20]. In this regard, it is well documented that the prevalence of CT exceeds 3% in women up to 40 years of age in other countries [21]. However, the presence of HR-HPV and STIs-O, as well as the rates of co-infection, remain insufficiently documented in our setting. Furthermore, a unifying criterion is currently being established at the national level to provide a standardised guideline for CC prevention programmes, which have previously been implemented at the discretion of each Autonomous Community [3,22]. Consequently, new local insights are essential to inform public health guidelines regarding screening, diagnosis, and treatment of HR-HPV infection and STIs-O, with the aim of reducing their contribution to the burden of disease.

Therefore, the present study aims to determine the presence of the major STIs (*Chlamydia trachomatis* (CT), *Trichomonas vaginalis* (TV), *Neisseria gonorrhoeae* (NG), and *Mycoplasma genitalium* (MG)) and the rate of co-infection with HR-HPV in women previously diagnosed with HR-HPV infection in a Spanish setting.

## 2. Materials and Methods

### 2.1. Study Setting and Design

A prospective cross-sectional observational study was conducted in the Miguel Servet University Hospital, a Spanish tertiary care hospital located in Zaragoza, which serves a total catchment area of over 400,000 people. Samples were collected from February 2024 to November 2024.

### 2.2. Participants and Sample Collection

All women attending the hospital’s Gynaecology Department for the CC screening programme, who met the eligibility criteria listed below and signed the informed consent form, were recruited to participate in this study. The following inclusion criteria were applied: participants had to be followed up for HR-HPV infection detected in the last year and be aged between 18 and 65 years old. As exclusion criteria, participants had to have no history of CC, and they could not have been treated for an STI-O or enrolled in an HPV vaccine study in the previous year.

Their endocervical specimens were collected by a trained gynaecologist and preserved in transport medium (PreservCyt^®^ solution, Hologic, MA, USA) for transfer to the laboratory on the same day. They were stored at room temperature until the next day, when they were analysed in accordance with the routine procedure.

### 2.3. Laboratory Testing and Outcomes of Interest

The samples were processed at the hospital microbiology laboratory for the detection of HR-HPV, and those with a positive result were subjected to further analysis to determine the presence of CT, NG, MG, and TV. All analyses were conducted using real-time PCR (polymerase chain reaction) on the cobas^®^ 5800 System with the use of cobas^®^ HPV, cobas^®^ CT/NG, and cobas^®^ TV/MG reagents (Roche Diagnostics, IN, USA).

The results were classified in accordance with the manufacturer’s specifications as either negative or positive. In cases where the initial classification was inconclusive, a second test was conducted to provide further clarification. In instances where uncertainty persisted, a new sample from the participant was analysed.

The results obtained from the cobas^®^ 5800 System software regarding the HR-HPV, CT, NG, TC, and MG tests were exported as a tab-separated values file from the instrument and converted into a spreadsheet for analysis. The designated investigator extracted the results for data storage and analysis, and the age of each participant at the time of sample collection was added as an additional outcome of interest.

### 2.4. Statistical Analysis

A descriptive analysis of the clinical results was conducted. The rates of STI co-infection were determined by calculating the ratio of the total number of cases to the number of positive results. Further analyses of the data were conducted using descriptive and inferential statistics. Categorical variables were presented as absolute and relative frequencies (n, %), and age was presented as the median and interquartile range. Group-wise comparisons by age were performed using Student’s *t*-test or Chi-square (χ^2^) test. A two-tailed *p* value was considered statistically significant when *p* ≤ 0.05. All analyses and graphs were conducted using the open-source software Jamovi 2.2.5, MS Excel 2016 (Microsoft, Redmond, WA, USA), MS PowerPoint 2016 (Microsoft, Redmond, WA, USA), and MS Word 2016 (Microsoft, Redmond, WA, USA).

### 2.5. Ethics

This study was conducted in accordance with the Declaration of Helsinki and its subsequent modifications. Prior to sample collection, all participants provided written informed consent. The protocol was approved by the Ethics Committee of Aragon and by the hospital management.

## 3. Results

A total of 254 women were selected for the present analysis, with a median age of 40.5 years (interquartile range 35.0–48.0). Appendix A shows that there were HR-HPV-positive cases included 23 (9.1%) HPV16, 12 (4.7%) HPV18, and 73 (28.7%) other HR-HPV cases; STIs-O-positive cases included 7 (2.8%) CT, 3 (1.2%) MG, 2 (0.8%) TV, and 1 (0.4%) NG cases.

Table 1 shows the overall infection rates, which were 38.6% for HR-HPV and 4.3% for ITSs-O (3.8% in HR-HPV-negative women and 5.1% in HR-HPV-positive women). Additionally, ITSs-O rates were shown according to age group. A higher rate of ITSs-O was found among women under 35 years of age with a HR-HPV positive result, in comparison to those with a negative HR-HPV result (20.0% vs. 4.2%). Furthermore, in this age group, the odds ratio for an ITS-O positive result in HR-HPV-positive women was 5.6 when compared to HR-HPV-negative women (*p* = 0.05). In the case of the eldest women, no STIs were found after 48 years of age (the positive ITS results found in those HR-HPV-negative women over the age of 45 belong to women aged between 47 and 48).

Appendix A illustrates the distribution of age according to the HR-HPV and ITS-O results. The distribution of age differs significantly according to the ITS-O result in both HR-HPV-positive and -negative women. No older women from 55 to 64 years of age tested positive for ITS-O (independently of the HR-HPV result), and only two (33%) women between 45 and 54 years tested positive for ITS-O among those with a negative HR-HPV result.

Appendix A depicts the values for age according to the presence or absence of each ITS. The age of women with a positive result for MG was found to be lower (31.0 (interquartile range 29.5–32.5)) than that of women with a negative result (41.0 (interquartile range 35.0–48.0)). Conversely, the age of women with a positive result for overall HR-HPV was higher (39.5 (interquartile range 33.0–47.3)) than that of women with a negative result (42.0 (interquartile range 37.0–48.8)). Both *p* values are ≤ 0.05.

Figure 1 and Table 2 shows the microorganisms involved in ITS co-infections. Figure 1 depicts a total of 104 infected specimens, representing 40.9% of the total number tested. Of these, 4 (1.6%) were triple infections, 9 (3.5%) were double infections, and 91 (35.8%) were single infections. HR-HPV (in particular those differing from HPV16 and HPV18) were the microorganisms most frequently involved (100% in triple and double infection and 93.4% in single infection), being followed by CT (present in 25.0%, 22.2%, and 4.4%, respectively) and MG (present in 50.0%, 0.0%, and 1.1%, respectively), as Table 2 shows. In addition, in women with HR-HPV positivity, one case was identified with MG and CT and another with MG and NG. The remaining cases were identified as co-infections between different types of HR-HPV. Eight specimens with multiple HR-HPV co-infections were found. These included five cases of co-infection with HPV16 and other HR-HPVs, one case of co-infection with HPV18 and other HR-HPVs, and two cases of co-infection with HPV16, HPV18, and other HR-HPVs. In samples from women negative for HR-HPV, only single ITS cases were found: CT was found in four samples, MG in one, and TV in another.

## 4. Discussion

In women participating in a CC screening program who had a previous HR-HVP positive result in the last year, the following rates were documented: a persistent positive result for HR-HPV (38.6%), an any-ITS rate (40.9%), and a multiple infection rate (5.1%) involving at least one HR-HPV (in 100%) and one ITS-O (in 38.5%). Furthermore, a significantly higher presence of ITS-O was observed in women under 35 years old with a persistent positive HR-HPV result when compared to women with a negative HR-HPV result (20% vs. 4.2%).

The present findings are in alignment with those of several studies from various countries, which also identified a simultaneous presence of HR-HPV with the STI-O [10,12,15,19,21]. However, it should be noted that the rates vary according to the specific population and methods under study. For instance, a study conducted in our city in 2005 found HR-HPV in the 8.4% of their total population, compared to our 38.6%, and this rate differed significantly according to age (10.6% vs. 24.2% in 25–35 years, 4.4% vs. 43.0% in 35–44 years, 5% vs. 45.8% in 45–54 years, and 0% vs. 39.4% in 55–64 years), but all these differences could be explained by the fact that their participants were not identified as having a previous HR-HPV positive result [23]. Likewise, a study on female sex workers in Galicia, published in 2012, found that HR-HPV presence was associated with bacterial vaginosis, including TV and *Candida* spp. [24]. Moreover, a large European study (with notable Spanish contributions) identified a positive correlation between the number of identified STIs and the risk for CC development [25]. To the best of our knowledge, however, the present study is the first to assess several STI-O regarding HR-VPH presence in our city and region.

Notwithstanding the unavailability of a framework for the comparison of studies, it is anticipated that our findings will contribute to the scientific evidence that the prevailing guidelines for CC screening should be observed to ensure their continued utility. Indeed, the Spanish CC screening programme is currently under review [26], and nowadays it includes a cytology every three years for those between 25 and 34 years of age and a HR-HPV test every five years for the rest under 65 years, except in case of a positive result for HR-HPV, for which cytology is conditionally performed and the frequency of HR-HPV testing is reduced to one year. Although Spanish guidelines already recognised that co-infection with some STI-O is one of the risk factors for VPH persistence and CIN and CC development [23], this concern is not currently incorporated in the general protocol, but it is asserted that specific action protocols should be implemented for women who meet the high-risk criteria for their individual risk assessment and follow-up [26].

In this regard, the findings of the present study suggest that the routine indication for ITS-O screening should be considered as conditional on a HR-HPV positive result for Spanish women, as it is aligned with the criteria described in the Spanish CC screening programme consensus [26], and it is evident that it would be cost-effective at the rates observed here [19,20]. In this context, it is important to note that the cost-effectiveness of STI screening varies according to the approach adopted [27]; however, most researchers agree that interventions should be justified based on the prevalence of STIs in each population, with a lower threshold of 3.1% suggested in a previous review of this assessment [20]. Furthermore, it is crucial to recognise that the cost-effectiveness of STI screening is often underestimated, as many studies fail to account for the long-term complications of STIs or the associated intersectoral or social costs, such as the impact on patients and their families, informal care, and work productivity [28]. Consequently, experts recommend implementing STI screening interventions at least in high-risk populations (where STI prevalence is higher), while also considering their cost-effectiveness in the general population, encompassing key aspects such as self-sampling, tailored strategies, and accessibility improvement [29]. Regarding this matter, self-sampling has been shown to reduce both costs and disparities in access to cervical swab testing, though professionals are still required to address the psychological impact of co-infection diagnoses and to establish effective treatments, especially given the stigma surrounding STIs, which may act as a barrier to healthcare access [30]. In response, experts advocate for reducing stigma by integrating sexual health services within a broader healthcare framework; this approach would also provide patients with opportunities for additional STI testing, including HIV screening, partner notification, and access to pre-exposure prophylaxis for HIV prevention (if appropriate), as well as other essential services [29]. Moreover, extending testing for STIs-O by using the same unique sample and extracting with a similar PCR technique by using multiplex kits (as enabled by Roche Diagnostic reagents, instruments, and platforms) would facilitate implementation into the laboratory routine contributing to improved cost-effectiveness.

In fact, given the imminent adoption of population-based CC screening in Spain in accordance with a European Commission directive [31], and the assertion that HR-HPV detection would serve as the most suitable primary screening test for vaccinated women (basically the younger women) due to its sensitivity and predictive value surpasses that of cytology [32], in which a higher frequency of STI detection is also observed for this population group, this particular scenario offers an opportunity for screening STI-O by PCR conditional on the primary HR-HPV result. This approach would be highly valuable for this population group, not only in terms of CC prevention but also in addressing some of the issues associated with STI-O testing in Spain [33].

However, it is important to recognise the limitations of this study. Its cross-sectional design does not allow for causality or other longitudinal effects to be observed. The potential for bias arising from the desire to participate could not be assessed. Moreover, the relatively small size of our sample limits the statistical power of our results and may give rise to type 1 error, particularly in the context of segmented analyses. However, this issue can be overcome by increasing the sample size through further recruitment, which is planned for the continuation of the present study. This will also allow for the identification of additional related findings.

Furthermore, it is evident that the local nature of this study means that findings should be interpreted with caution when attempting to extrapolate them to other populations or healthcare settings, in which the presence of potential confounders (e.g., sexual behaviour, vaccination status, immunosuppression, smoking, alcohol consumption, or socio-economic status) may impact both HR-HPV and STIs-O rates in different ways [34]. In this sense, it is noteworthy to acknowledge that the incidence and prevalence rates of cervical cancer in Zaragoza are consistent with those of Spain (5 and 31 per 100,000 inhabitants, respectively) [1] and are among the lowest in the European Union (EU) [22]. Comparing with the EU rates, the Spanish CC screening participation is lower (access disparities related to education and income groups are present), and the proportion of daily smokers and individuals with excess of weight is higher; however, alcohol consumption is lower, and HPV vaccination among 15-year-old girls is among the highest [22]. Furthermore, the employment of different diagnostic methods used for identifying STIs or microorganisms may potentially lead to misclassification of infection status, depending on their level of sensitivity and specificity [35].

On the other hand, the present study focuses on the major STIs, in accordance with the research priorities of the WHO 2022–2030 agenda for STI diagnosis and prevention [36]. However, cervicovaginal dysbiosis, characterised by a decrease in lactobacilli populations and an increase in both alpha and beta diversity, has also been found to be associated with different stages of CC development and other related gynaecological conditions, involving additional microorganism species through mechanisms and relationships that are beginning to be elucidated [2,34,37,38,39,40]: bacteria such as *Gardnerella vaginalis*, *Atopobium vaginae*, *Sneathia amnii*, *Lactobacillus iners*, *Prevotella* spp., and *Ureaplasma* spp., fungus such as *Sporidiobolaceae*, *Saccharomyces*, *Candida* spp., and *Malassezia*, and viruses such as herpes simplex 2 and anelloviruses. Likewise, several host endogenous risk factors for CC, such as nutritional or genetic features, could be considered for the management of CC prevention [34]. Therefore, further studies are required to explore the usefulness of additional microorganism determinations and other biomarkers in order to improve future preventive screening.

## 5. Conclusions

The findings of the present study support the benefits of implementing STI screening in Spanish women under 50 years old previously diagnosed with a HR-HPV infection, especially in those under 35.

Given the potential benefits and projected cost-effectiveness of incorporating STI-O testing into preventive CC screening, policymakers should consider implementing evidence-based interventions in this manner, particularly among high-risk and younger populations. Further research is encouraged to identify and optimise the most effective approaches.

## Figures and Tables

**Figure 1 viruses-17-00247-f001:**
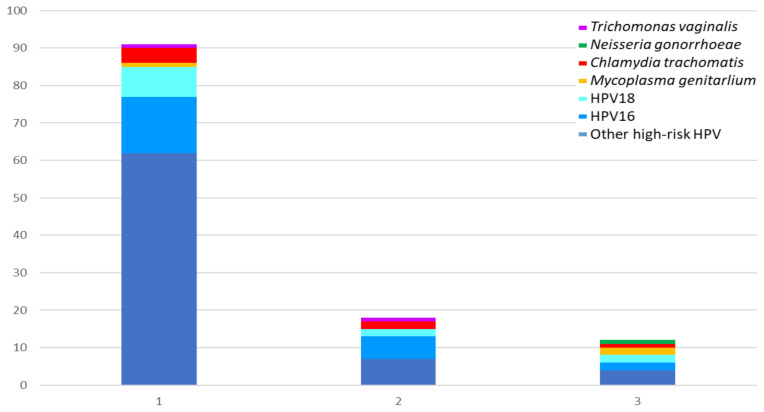
ITS rate by number of co-infections (single, double, and triple).

**Table 1 viruses-17-00247-t001:** STIs rates by HR-HPV test result.

	HR-HPV-Positive	STI-O in HR-HPV-Positive 98/254 (38.6%)	STI-O in HR-HPV-Negative 156/254 (61.4%)	Raw OR (CI 95%)	*p* Value
Overall	98/254 (38.6%)	5/98 (5.1%)	6/156 (3.8%)	1.34 (0.4–4.5)	0.63
Age (years)					
25–34	15 (24.2%)	3/15 (20.0%)	2/47 (4.2%)	5.6 (0.8–37.6)	0.05
35–44	43 (43.0%)	2/43 (4.6%)	2/57 (3.5%)	1.3 (0.2–9.9)	0.78
45–54	27 (45.8%)	0/27 (0.0%)	2/32 (6.2%)	0.22 (0.0–4.8)	0.19
55–65	13 (39.4%)	0/13 (0.0%)	0/20 (0.0%)	-	-

Note. HR-HPV: high-risk human papillomavirus, STI-O: sexually transmitted infection other than HR-HPV, OR: odds ratio.

**Table 2 viruses-17-00247-t002:** Major microorganisms involved in double and triple STIs co-infection.

No. of Involved Microorganisms	*Chlamydia trachomatis*	*Neisseria gonorrhoeae*	*Mycoplasma genitalium*	*Trichomonas vaginalis*	HPV-16	HPV-18	Other HR-HPV	No. HR-HPV
3	0	0	0	0	1	1	1	3
3	0	0	0	0	1	1	1	3
3	1	0	1	0	0	0	1	1
3	0	1	1	0	0	0	1	1
2	0	0	0	0	1	0	1	2
2	0	0	0	0	1	0	1	2
2	0	0	0	0	1	0	1	2
2	0	0	0	0	1	0	1	2
2	0	0	0	0	0	1	1	2
2	0	0	0	0	1	0	1	2
2	1	0	0	0	0	0	1	1
2	1	0	0	0	0	1	0	1
2	0	0	0	1	1	0	0	1

Note. HR-HPV: high-risk human papillomavirus, HPV: human papillomavirus, No.: number, and STIs: sexually transmitted infections.

## Data Availability

The original contributions presented in this study are included in this article/Appendix A. An anonymised raw dataset is openly available as a Appendix A. Further inquiries can be directed to the corresponding author.

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
