# Peer review of "HPV-Associated Sexually Transmitted Infections in Cervical Cancer Screening: A Prospective Cohort Study"

_viruses, 2025, doi:10.3390/v17020247_

Round 1
Reviewer 1 Report
Comments and Suggestions for Authors
This is an interesting manuscript that necessitates certain improvement before it can be accepted for publication. While the study included 254 participants, this is relatively small for generating generalizable results, especially when divided into subgroups by age and infection status. This limitation is noted but not adequately addressed in terms of its impact on statistical power. The authors fail to address how these findings might be extrapolated to other regions or healthcare settings. The study does not adequately discuss potential confounders, such as sexual behavior, socio-economic status or vaccination status, which could impact both HR-HPV and STI rates. Also, the diagnostic methods used for identifying STIs or microorganisms may have varying levels of sensitivity and specificity, potentially leading to misclassification of infection status, which should also be accounted for in the Limitations section of the manuscript.
The focus on only four STIs (Chlamydia trachomatis, Neisseria gonorrhoeae, Mycoplasma genitalium and Trichomonas vaginalis) neglects other potential contributors to cervical cancer development, such as Ureaplasma spp., Candida spp., bacterial vaginosis pathogens, etc.. This is noted among limitations, but it should be discussed in depth in the Discussion section of the manuscript. Also, more references tackling the same topic from 2024 and 2025 should be included.
While the study suggests incorporating STI screening into cervical cancer programs, it does not address how this might practically affect public health strategies or healthcare costs. This warrants a much larger discussion. Also, the cost-effectiveness analysis is mentioned briefly but lacks depth. The study also does not explore the potential social or psychological impact of co-infection diagnoses, especially given the stigmatization of STIs; therefore, addressing these aspects would add depth to the discussion.
In the Introduction section of the manuscript, in lines 56-57, it is stated “Indeed, screening for CT is recommended in sexually active women under 25 years of age and in sexually active women over 25 years old when at increased risk”, but it is not clear what increased risk actually pertains to. Also, when mentioning complications, more emphasis should be on pelvic inflammatory disease. Conclusion section should be broadened with more policy-relevant recommendations.
Author Response
Dear reviewer,
Thank you very much for your review.
Please find attached our reply.

Reviewer 2 Report
Comments and Suggestions for Authors
The study is straightforward, and the result is clearly interpreted. May I suggest the authors indicate the number of HPV16 and HPV18 cases in the result. In L141-142, "the odds ratio.......was 5.2...", but it is not indicated in Table 1.
Author Response

(The authors gave the same response as above.)

Reviewer 3 Report
Comments and Suggestions for Authors
The research investigates co-infections with HPV, an understudied aspect of HPV and cervical cancer. As in Introduction section explains, previous research has identified some potentially synergistic effects between some species of bacteria and HPV. Figure 1 had readable axes labels and colour legend. A couple of minor issues:
1. Brand names and trademarks of the instruments and kits are not suitable for Abstract section. Keep them to the Methods section.
2. Discussion section could have a couple of sentences about future research work.
Author Response

(The authors gave the same response as above.)
